Novel Systems Biology Techniques

# MICOM: Metagenome-Scale Modeling To Infer Metabolic Interactions in the Gut Microbiota

Christian Diener,[a,b] Sean M. Gibbons,[b,c] Osbaldo Resendis-Antonio[a,d]

[a]Instituto Nacional de Medicina Genómica (INMEGEN), Mexico City, México
[b]Institute for Systems Biology, Seattle, Washington, USA
[c]eScience Institute, University of Washington, Seattle, Washington, USA
[d]Human Systems Biology Laboratory, Coordinación de la Investigación Científica - Red de Apoyo a la Investigación, Universidad Nacional Autonóma de México (UNAM), Mexico City, México

**ABSTRACT** Compositional changes in the gut microbiota have been associated with a variety of medical conditions such as obesity, Crohn's disease, and diabetes. However, connecting microbial community composition to ecosystem function remains a challenge. Here, we introduce MICOM, a customizable metabolic model of the human gut microbiome. By using a heuristic optimization approach based on L2 regularization, we were able to obtain a unique set of realistic growth rates that corresponded well with observed replication rates. We integrated adjustable dietary and taxon abundance constraints to generate personalized metabolic models for individual metagenomic samples. We applied MICOM to a balanced cohort of metagenomes from 186 people, including a metabolically healthy population and individuals with type 1 and type 2 diabetes. Model results showed that individual bacterial genera maintained conserved niche structures across humans, while the community-level production of short-chain fatty acids (SCFAs) was heterogeneous and highly individual specific. Model output revealed complex cross-feeding interactions that would be difficult to measure *in vivo*. Metabolic interaction networks differed somewhat consistently between healthy and diabetic subjects. In particular, MICOM predicted reduced butyrate and propionate production in a diabetic cohort, with restoration of SCFA production profiles found in healthy subjects following metformin treatment. Overall, we found that changes in diet or taxon abundances have highly personalized effects. We believe MICOM can serve as a useful tool for generating mechanistic hypotheses for how diet and microbiome composition influence community function. All methods are implemented in an open-source Python package, which is available at https://github.com/micom-dev/micom.

**IMPORTANCE** The bacterial communities that live within the human gut have been linked to health and disease. However, we are still just beginning to understand how those bacteria interact and what potential interventions to our gut microbiome can make us healthier. Here, we present a mathematical modeling framework (named MICOM) that can recapitulate the growth rates of diverse bacterial species in the gut and can simulate metabolic interactions within microbial communities. We show that MICOM can unravel the ecological rules that shape the microbial landscape in our gut and that a given dietary or probiotic intervention can have widely different effects in different people.

**KEYWORDS** flux balance analysis, gut microbiome, metagenome, systems biology

The composition of the gut microbiome can influence host metabolism (1) and has been associated with a variety of health conditions such as obesity, Crohn's disease, diabetes, and colorectal cancer (2–6). However, the causal roles played by the gut microbiota in host physiology and disease remain unclear. Several studies have

This article followed an open peer review process. The review history can be read here.

Address correspondence to Sean M. Gibbons, sgibbons@isbscience.org, or Osbaldo Resendis-Antonio, oresendis@inmegen.gob.mx.

MICOM, a metagenome-scale metabolic modeling package, can reproduce the diverse growth rates of bacteria in the gut, predict the metabolic exchanges that drive the microbial landscape, and suggest optimal interventions.

mapped individual gut microbial genes to functions (7–9). However, these mappings are largely qualitative, as the presence of a particular gene does not guarantee expression of a functional enzyme. An alternative strategy to quantify the metabolic capacity of a microbial community is to use computational models for inferring fluxes in biochemical networks (10, 11). While direct experimental measurement of fluxes by carbon or nitrogen labeling is costly, one can readily estimate the metabolic fluxes of a model organism using genome-scale metabolic models. For individual bacteria, metabolic modeling using flux balance analysis (FBA) has been a valuable tool for exploring metabolic capacities under different conditions and has been used extensively in basic research, biochemical strain engineering, and *in vitro* models of bacterial interactions (12–15). In FBA, fluxes are usually approximated from a genome-scale model containing all known biochemical reactions by maximizing the production of biomass under constraints mirroring enzymatic, thermodynamic, and environmental conditions (13). For instance, one can restrict metabolic import fluxes to those whose substrates are present in the growth medium (12, 14, 16) in order to simulate a particular nutrient environment. Extending FBA to microbial communities is challenging due to the necessity of modeling metabolic exchanges between many taxa and selecting an appropriate objective function to account for potential trade-offs between species and community growth rates.

Maximizing the community growth rate is at odds with maximizing individual species growth rates. Multiobjective methods, like OptCom, attempt to find the joint maximum of individual and community growth rates (17). However, these multiobjective methods are limited to smaller-sized communities. The human gut microbiome, on the other hand, may contain up to several hundred distinct species (18). An additional challenge is the integration of relative abundances obtained from 16S amplicon or metagenomic shotgun sequencing into a community FBA model. This is particularly important for accurately inferring the metabolic exchanges taking place between different species within the community. A very abundant species should import and export much greater absolute quantities of metabolites than a very rare species, which in turn impacts the resulting community-level biochemical fluxes. Despite the challenges, genome-scale metabolic modeling of microbial communities holds great promise as a tool for estimating the metabolic potential of an individual's gut microbiome. In particular, this approach could yield valuable insights into possible metabolic mechanisms underlying host disease states.

Here, we present a computational approach that efficiently extends metabolic modeling to entire microbial communities. Using a two-step optimization procedure, we were able to simulate growth and metabolic exchange fluxes for metagenome-scale metabolic models of ecologically diverse bacterial systems. Additionally, we explicitly included microbial abundance estimates from metagenomic shotgun sequencing and realistic dietary inputs in order to make quantitative, personalized, metabolic predictions. This entire strategy is implemented in an open-source Python software package called MICOM (shorthand for *mi*crobial *com*munity).

We tested our approach by applying MICOM to a balanced data set of 186 Danish and Swedish individuals, including healthy controls, patients with type 1 diabetes, and patients with type 2 diabetes with and without metformin treatment. We show that individual bacterial growth rates vary greatly across samples and are correlated with independently measured replication rates. We quantified exchanges between the gut microbiota and gut lumen and studied the effect of the microbiota composition on the production of short-chain fatty acids (SCFAs) for samples from healthy and diabetic individuals. Overall, we found that MICOM predicted a bimodal usage pattern of dietary metabolites, ecological interactions between microbes tended to be community specific and largely competitive, key gut genera associated with health participated in the largest number of ecological interactions, inferred SCFA production was lower in diabetic patients, and targeted dietary or probiotic interventions had unique functional consequences for each individual.

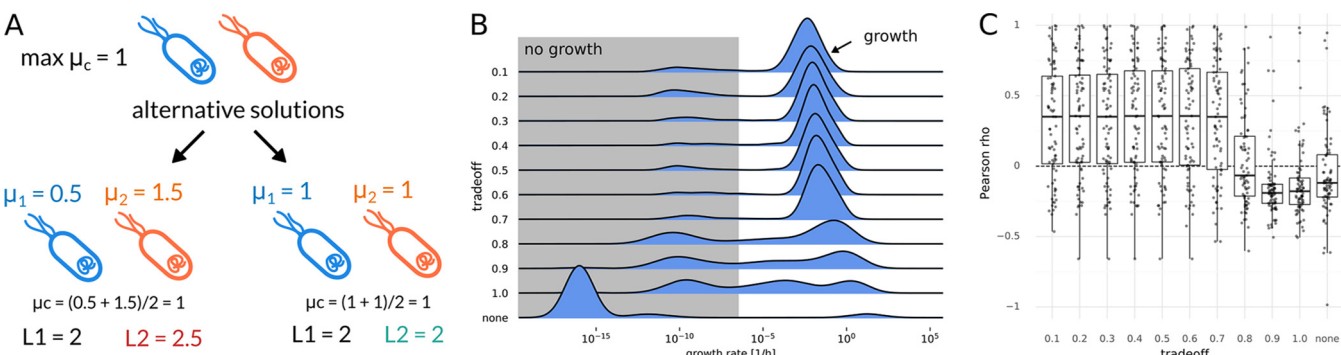

**FIG 1** Regularization of growth rates. (A) Regularization values for a toy model of two identical *Escherichia coli* populations. Two alternative solutions are shown with different individual growth rates and respective values of regularization optima. Here, L1 denotes minimizing the sum of growth rates, whereas L2 denotes minimizing the sum of squared growth rates. Only L2 regularization favors one over the other and identifies the expected solution where both populations grow with the same rate. (B) Effects of different trade-off values (fraction of maximum community growth rate) on the distribution of individual genus growth rates. Zero growth rates were assigned a value of $10^{-16}$, which was smaller than the observed nonzero minimum. Growth rates smaller than $10^{-6}$ were considered to not represent growth (gray shaded area). (C) Pearson correlation between replication rates and inferred growth rates with different trade-off values. "none" indicates a model without regularization returning arbitrary alternative solutions (see Materials and Methods). The dashed line indicates a correlation coefficient of zero.

## RESULTS

**A regularization strategy for microbial community models.** Metabolic modeling is commonly applied to model a single strain of bacteria in log phase, where the growth rate is approximately constant and the log of the bacterial abundance increases linearly with time. Modeling bacterial growth in natural environments is often more complex than this, but some information on environmental context can be extracted from the relative abundances of bacterial taxa. Within a single individual and in the absence of persistent dietary changes, gut microbial relative abundances tend to fluctuate around a fixed median value over month-to-year time scales (19–21). This is consistent with a steady-state model where bacterial growth is in equilibrium with a dilution process that continuously removes biomass from the system (22). Using this approximation, bacterial growth rates are constants $\mu_i$ (in 1/hour), which is compatible with the assumption of FBA. All bacteria in the microbial community then contribute to the production of total biomass, with an overall growth rate constant $\mu_c$. The community growth rate $\mu_c$ is obtained from the individual growth rates $\mu_i$ by a weighted mean, with the relative contribution of species i ($a_i$) to the total biomass serving as the weight (17, 22).

$$\mu_C = \sum_i a_i \mu_i \qquad (1)$$

Even though FBA can be used to obtain the maximum community growth rate, one can see from equation 1 that there is an infinite combination of different individual growth rates $\mu_i$ for any given community growth rate $\mu_c$ (see Fig. 1A for an example). Various strategies have been employed in order to deal with this limitation, where the simplest strategy is to report any one of the possible growth rates distributions for $\mu_i$. Other approaches attempt to find the set of growth rates that maximize community growth and individual growth at the same time (17), but this is computationally intensive and may not scale well to the species-diverse gut microbiome (18, 23). Thus, we formulated a strategy that allows us to identify a realistic set of individual growth rates $\mu_i$ and scales to large communities. The simplest case of a microbial community is a community composed of two identical clonal strains, each present in the same abundance. Assuming that the maximum community and individual growth rates are equal to 1.0, there are now many alternative solutions giving maximal community growth (Fig. 1A). However, the two populations are identical and present in the same abundance, so one would expect that both grow at the same rate. In order to enforce a particular distribution of individual growth rates, one can try to optimize an additional function over the individual growth rates $\mu_i$. This is known as regularization, and a feasible regularization function should enrich for biologically relevant growth rate

distributions. As a heuristic, our minimal requirement for a feasible regularization function was consistency with the observed metagenomic abundances. This means that a taxon that is observed in the data should be able to grow. Thus, the growth rate of a taxon should be nonzero if its abundance is nonzero. We show in Text S1 in the supplemental material that no linear regularization function can comply with that requirement, whereas a simple quadratic regularization, also known as L2 regularization, does fulfill that requirement (24, 25). L2 regularization is known to distribute magnitude over all variables, which is also consistent with a maximization of individual growth rates and thus forms a heuristic for the simultaneous maximization of individual and community growth rates.

L2 regularization can be readily integrated into FBA as a quadratic optimization problem, which is not necessarily true for any generic function. In the previous example of two identical strains, only the L2 norm correctly identifies the solution where both strains grow at the same rate as optimal. Additionally, the L2 norm has a unique minimum. Thus, there is only one configuration of individual growth rates $\mu_i$ that minimizes the L2 norm for a given community growth rate $\mu_c$. In practice, maximal community growth might be achievable only if many taxa are excluded from growth, for instance by giving all resources to a fast-growing subpopulation. Again, this is inconsistent with reality if one has prior knowledge that the other taxa are present in the gut and should be able to grow. Instead of enforcing the maximal community growth rate, one can limit community growth to only a fraction of its maximum rate, thus creating a trade-off between optimal community growth and individual growth rate maximization. Community growth maximization requires full cooperativity, whereas the L2 norm minimization represents selfish individual growth maximization. Thus, we call our two-step strategy of first fixing community growth rate to a fraction of its optimum and then minimizing the L2 norm of individual growth rates a "cooperative trade-off." Even though it is difficult to formulate a closed form solution for this two-step optimization, one can obtain a solution for the second optimization (minimization of regularization term) when dropping additional constraints for growth rates (see Text S1 for derivation). In that case, growth rates are given by:

$$\mu_i = \frac{\alpha \mu_c}{a^T a} a_i \qquad (2)$$

Thus, optimal growth rates will be approximately correlated with abundance where the slope depends on the abundance distribution and the maximum community growth rate.

We found that computation time generally scaled well with the community size (with most individual optimizations taking less than 5 min) when using interior point methods, which are known to provide better performance for larger models (26). However, we found that it was difficult to maintain numerical stability with large community models. None of the tested solvers were able to converge to optimality when solving the quadratic programming problem posed by the L2 norm minimization (see Materials and Methods). Thus, we used a crossover strategy to identify an optimal solution to the L2 minimization (see Materials and Methods).

**Regularization by cooperative trade-off yields realistic growth rate estimates.** In order to test whether cooperative trade-off yields realistic growth rates, we implemented and applied it to a set of 186 metagenome samples from Swedish and Danish individuals (27), consisting of healthy individuals, individuals with type 1 diabetes, and individuals with type 2 diabetes stratified by metformin treatment (a known modulator of the gut microbiome) (28). Relative abundances and cleaned coverage profiles for a total of 239 bacterial genera and 637 species were obtained with SLIMM (29) from previously published metagenomic reads (27, 29) as described in Materials and Methods. We used ratios in coverage between replication initiator and terminus as a measure for replication rates, which have been reported to be good proxies for bacterial growth rates *in vivo* (30). This provided a set of 1,571 strain-level replication rate measurements for the 186 samples that were used for validation of the inferred

mSystems®

**TABLE 1** Distribution of taxon assignments for ranks

| Taxon | No. of unique taxa[a] | % of reads[b] | |
| --- | --- | --- | --- |
| | | Assigned reads[c] | With model[d] |
| Kingdom | 1 | 100 ± 0 | 100 ± 0 |
| Phylum | 22 | 100 ± 0 | 99 ± 0 |
| Class | 32 | 100 ± 0 | 99 ± 0 |
| Family | 102 | 100 ± 0 | 91 ± 0 |
| Genus | 239 | 94 ± 5 | 91 ± 5 |
| Species | 637 | 79 ± 9 | 52 ± 9 |

[a]Number of unique taxa for each rank.
[b]Percentages of reads are shown as means ± standard deviations for the 186 samples. Only reads classified as bacteria were considered.
[c]Percentage of mapped reads that could be uniquely assigned to taxa within the rank.
[d]Percentage of reads whose taxon had at least one representative in the AGORA genome-scale metabolic models.

growth rates (1,062 and 1,113 on the genus and species levels, respectively; see Materials and Methods). Abundance profiles for all identified genera for all samples were connected with the AGORA models, a set of manually curated metabolic models which currently comprises 818 bacterial species (31). Ninety-three gut-associated genera within the AGORA reconstructions (version 1.03) represented more than 91% ± 5% of metagenomic reads for the 186 samples (see Table 1, genus row). Even though the cooperative trade-off strategy is applicable to species- or even strain-level data, the AGORA reconstructions accounted for only 52% ± 9% of all bacterial species in the data set. Thus, we decided to perform community model construction separately on the species level as well as the genus level, which covered a larger fraction of the observed microbiome. To accomplish this, individual strain models from AGORA were pooled into the higher phylogenetic ranks (see Materials and Methods). After removing low-abundance taxa (<0.1% for genera and <0.01% for species), the resulting communities contained between 12 and 30 taxa at the genus level and between 23 and 81 taxa at the species level. Each taxon was represented by a full genome-scale metabolic model and connected by exchange reactions with the gut lumen, thus yielding two sets of 186 complete metagenome-scale metabolic models (one set for the species level and one for the genus level). We used the relative read abundances as a proxy for the relative biomass of each taxon in each sample (see Materials and Methods). Even though relative abundances from shotgun metagenomes are not an exact representation of bacterial mass (in grams [dry weight]), we argue that the discrepancy between the two is probably much smaller than the variation in intertaxon abundances, which spans several orders of magnitude (18).

The data on 186 individuals used in this analysis did not include diet, metabolomics, or data on total microbial load. Thus, we were limited to study metabolic effects that are driven by microbiota composition alone and not by additional factors such as diet or total bacterial biomass. To use a moderately realistic set of import constraints for the community models, we modeled all individuals as consuming an average Western diet (32). Import fluxes for external metabolites were based on a reported set of fluxes for an average Western diet (31, 33). To account for uptake in the small intestine, we reduced all import fluxes for metabolites commonly absorbed in the small intestine by a factor of 10 (34).

To evaluate the performance of the cooperative trade-off, we compared the inferred growth rates with the replication rates obtained directly from sequencing data (see Materials and Methods). First, to establish a baseline, we ran an optimization that maximized only the community growth rate and used the distribution of growth rates returned by the solver when applying no regularization. This was followed by applying the cooperative trade-off strategy with various levels of suboptimality ranging from 10% to 100% of the maximum community growth rate. The predicted growth rates were now compared to the mean measured replication rates for each taxon in each sample using Pearson correlation (see Materials and Methods). As stated above, we

observed that simply optimizing the community growth rate with no regularization of the individual growth rates led to solutions where only a few taxa grew with unreasonably high growth rates (doubling times shorter than 5 min), whereas the rest of the microbial community had growth rates near zero (compare Fig. 1B with tradeoff of "none"). Consequently, the resulting model growth rates were uncorrelated with replication rates (mean Pearson rho = −0.02). Adding the L2 norm minimization while maintaining maximum community growth allowed more genera to grow (see Fig. S1 in the supplemental material) but yielded growth rates that were anticorrelated with replication rates (mean $r$ = −0.11). Lowering the community growth rate to suboptimal levels strongly increased the growing fraction of the population (Fig. S1) and led to a much better agreement with replication rates for trade-off values smaller than 0.7 (mean Pearson rho $\cong$ 0.4). Calculating correlations for all samples rather than within samples showed a similar tendency, with no regularization showing no correlation with replication rates ($r$ = −0.05, Pearson exact test $P$ = 0.07) and increased agreement up to a trade-off value of 0.5 ($r$ = 0.21, Pearson exact test $P$ = 2e−12). The lower magnitude correlations in the across-sample setting is likely due to differences in diet or bacterial load for people that were not taken into account. Overall, the best agreement with the observed replication rates across and within samples was observed at a trade-off value of 0.5. Using Spearman correlation instead of Pearson correlation also identified the same optimal trade-off, albeit with a slightly lower mean correlation within samples (Spearman $r$ = 0.28) and slightly stronger correlation for samples (Spearman $r$ = 0.27, Spearman exact test $P$ = 7e−19). Thus, the observed Pearson correlation results were not dominated by outliers. We also observed similar performance with the species-level models (Fig. S2). For the Spearman analysis, the best agreement with *in vivo* replication rates was observed for a trade-off parameter of 0.7 (Fig. S2C). Because the genus-level models performed equally well as the species-level models but represented a higher percentage of observed reads (Table 1), we decided to continue all further analyses using genus models with a trade-off parameter of 0.5.

**Growth rates are heterogeneous and depend on community composition.** A trade-off value of 0.5 for maximal community growth led to good agreement with replication rates. Bacterial communities showed an average doubling time of about 6 h, where individual genera had an average doubling time of 11 h. Community growth did not vary substantially for samples (0.246 ± 0.002 1/h), indicating that each individual's microbiota was almost equally efficient at converting dietary metabolites into biomass at the community level. However, we found that individual genus-level growth rates often varied over 5 orders of magnitude (Fig. 2). *Bacteroides* was predicted to be the fastest growing genus overall and was closely followed by *Eubacterium*, which is consistent with the ubiquitous presence of these abundant taxa in microbiome samples (35, 36).

In the absence of additional constraints, L2 regularization will result in growth rates that are linearly dependent on the taxon abundances (see equation 2 and Text S1). However, this requires some simplifying assumptions that may not be met in the particular constraints of the full metabolic community models. We compared growth rate estimates obtained from numerical optimization with the approximation from equation 2. We found that growth rates obtained with the cooperative trade-off usually followed the derived linear relationship, albeit with a large variation (mean $R^2$ = 0.94, standard deviation [SD] = 0.34; Fig. 3A). Deviations from that relationship were mostly observed for small growth rates (Fig. 3A) which could not reach the suggested growth rate due to additional constraints on growth. Thus, the linear relationship between growth rates and abundance holds for most growth rates but is likely inaccurate for very small growth rates. It is important to note that even though abundances are positively correlated with growth rates within a single individual, this is not true across samples where one can observe a negative correlation for abundant taxa (Fig. 3A). This is a consequence of the coefficient in equation 2, which depends on the actual abundance distribution as well. In particular, the slope of the linear relationship

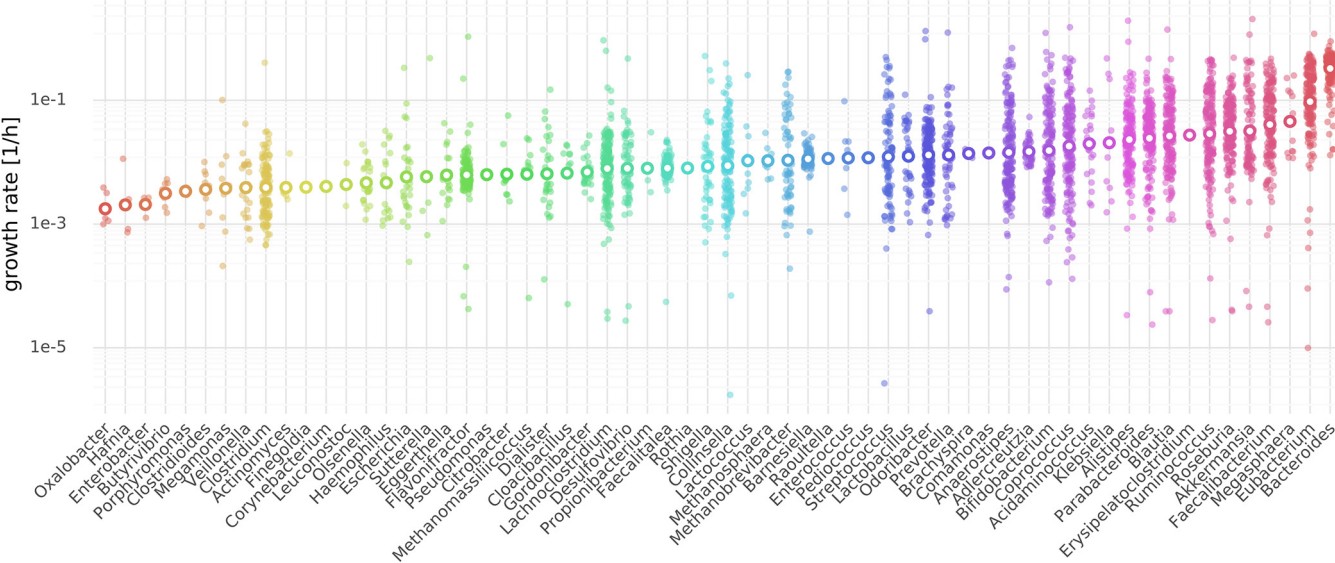

**FIG 2** Nonzero growth rates ($>10^{-6}$) for genera obtained by cooperative trade-off (50% maximum community growth rate). Each small solid circle denotes a growth rate for one sample of the 186 samples, and larger circles with white centers denote the mean growth rate for the genus (see Materials and Methods). Genera are sorted by mean growth rate from lowest growth rate (left) to the highest growth rate (right).

between abundance and growth rate will be the greatest if all taxa have equal abundances and take its lowest value when one taxon dominates.

We observed a wide variation in individual taxon growth rates for samples. Because all of the community models were constrained by the same diet, this phenomenon was due to microbiota composition only. To explain this variation in individual growth rates, we hypothesized that different genera might influence each other's growth rate, either

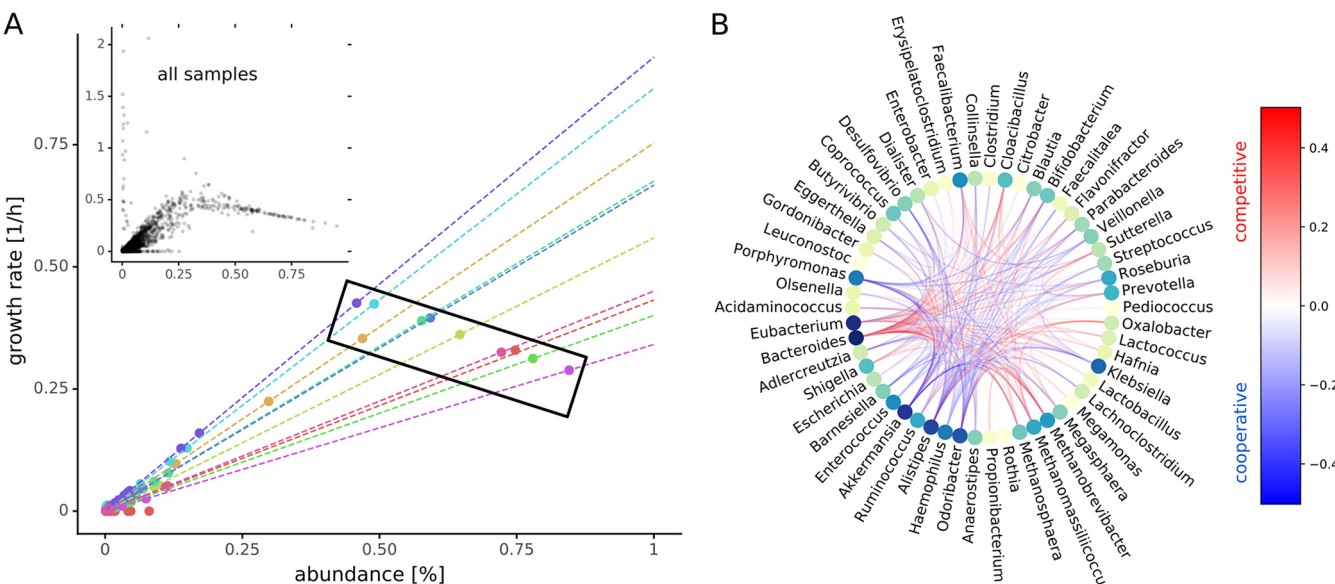

**FIG 3** Codependencies of growth rates. (A) Relationship between abundance and growth rate for samples. The large scatter plot shows growth rates and abundances for the first 10 samples. Each circle denotes one genus in one sample and its sample provenance is indicated by the color. Dashed lines denote the linear relationship between growth rates and abundances predicted by equation 2 for each sample. The black box demonstrates how different slopes (i.e., as community evenness declines, so does the within-sample slope) can result in negative correlation between abundance and growth rate across the samples. The smaller inset scatter plot shows data from all samples (Pearson rho = 0.69, $n$ = 39,815). (B) Growth rate interactions between genera as estimated by genus knockouts. Only interactions that induce a mean growth rate change of 0.1 for all samples (i.e., ubiquitous interactions) are shown. The color of the edges indicates change of growth rate and type of interaction. Red edges denote competition where removal of one genus increases the growth rate of the other, and blue edges denote cooperation or syntropy where the removal of one genus lowers the growth rate of the other. The nodes are colored by the degree (number of total connections) from lime (few) to dark blue (many).

mSystems®

by competition or by cooperation. In order to quantify growth rate interdependencies, we performed *in silico* knockouts for each genus in each sample and tracked the change in growth rates for all remaining genera in the sample (see Materials and Methods). Here, we found that the growth rate of each genus was influenced by another genus in at least one of the 186 samples. As would be expected for bacterial species competing for the same resources, most interactions were competitive (red edges in Fig. 3B). However, we observed a distinct subset of bacteria that were interconnected by a network of cooperative interactions, including *Akkermansia* and *Faecalibacterium* (blue edges in Fig. 3B; see also Fig. S3). Strikingly, genera participating in many interactions for all samples, such as *Bacteroides*, *Eubacterium*, *Akkermansia*, *Alistipes,* and *Faecalibacterium*, are known to be ubiquitous members of the gut microbiome and are often associated with health (6, 37–41). We found that the prevalence and strength of interactions were highly dependent on the composition of the microbiome. The vast majority of strong growth interactions were present in only one-to-five samples, whereas all other samples showed very few strong interactions (Fig. S3). This result is perhaps unsurprising, as many strong species-species interactions are thought to be destabilizing to ecological communities (42, 43).

**Analysis of exchange fluxes reveals differential use of diet components and niche partitioning in the microbiota.** One of the major modes of interaction between the gut microbiota and the host is by means of consumption or production of metabolites in the gut. In our simulations, all individuals were on the same average Western diet (see Materials and Methods), which imposes an upper bound for the flux of metabolites into the gut lumen. However, this does not determine *a priori* which components of the diet are consumed at what rate in each sample, because individual microbiota may consume less than what is imposed by that maximum diet flux. We quantified this effect by obtaining all import and export fluxes for each individual genus for all samples (1,613 exchange reactions in each of 62 genera) as well as metabolite exchanges between the microbiota and the gut lumen (152 metabolites). This was done in the absence of a metabolic model for enterocytes, colonocytes, or goblet cells due to the lack of a curated metabolic reconstruction and validated objective function for those cells. A unique set of exchange fluxes was obtained by considering the set of exchange fluxes with smallest total import flux for the growth rates obtained by the cooperative trade-off (see Materials and Methods). This assumes that the microbiota compete for resources with the host gut and will thus favor an efficient import that yields the maximum growth rate. This also corresponds to the particular distribution of import fluxes an individual microbiota is most adapted to.

Even though the minimization of total import fluxes favors simpler medium compositions, most samples showed a diverse consumption of metabolites from the gut, particularly in the wide array of carbon and nitrogen sources (Fig. 4A). There was a large set of metabolites that were consumed for all samples, but we also observed many metabolites with differential import fluxes for individuals. In particular, we observed a bimodal distribution where microbiota either consumed fibers and starches or branched-chain amino acids (see the metabolites indicated in Fig. 4A). This bimodal pattern did not correlate with health or disease states. As expected, all communities showed net anaerobic growth. Those preferred usage patterns did not seem to depend on the choice of the dilution factor used to lower import flux bounds for metabolites that are commonly absorbed in the small intestine (Fig. S4).

Given the observed heterogeneity in taxon growth rates and the large number of interactions, we were interested in looking at the uptake rates of metabolites from the gut lumen for each genus. There was overlap of metabolite usage across genera. On average 32% of the metabolites were shared between any two genera in any sample (standard deviation of 13%; see Materials and Methods). Consequently, more than two thirds of metabolites were used differentially between pairs of genera. To visualize the structure of metabolite consumption by individual bacterial genera in the gut, we used t-distributed stochastic neighbor embedding (t-SNE) dimensionality reduction on genus-specific import fluxes (44). This revealed clear genus-specific niche structure for

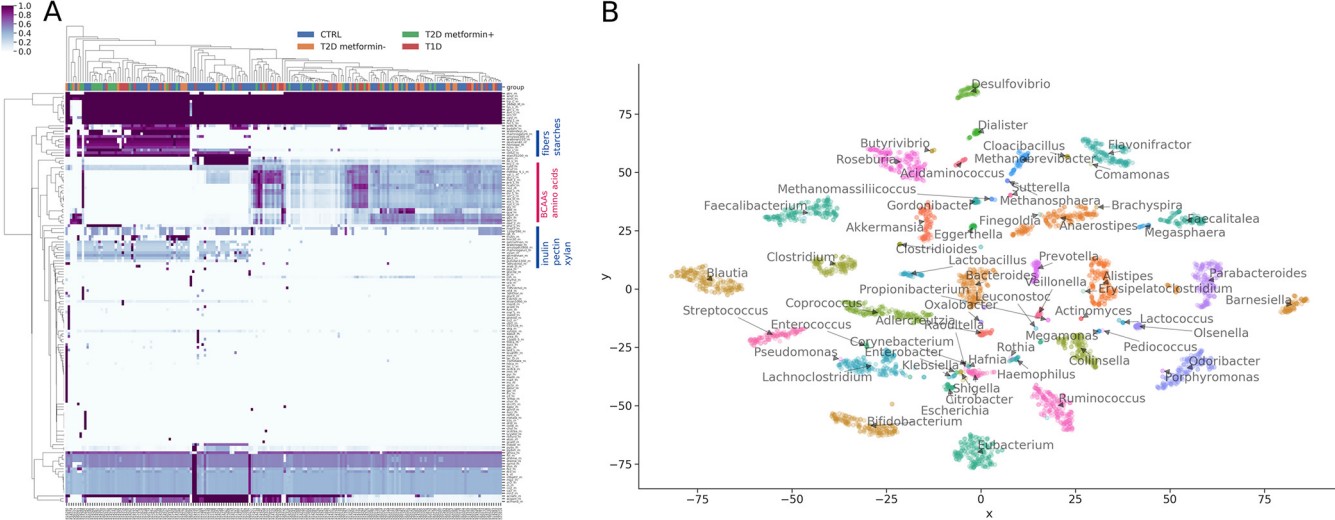

**FIG 4** Microbiota import fluxes across samples. Exchange fluxes were calculated as the smallest set of import fluxes that could maintain the genera growth rates obtained by the cooperative trade-off. (A) Import fluxes for samples. Rows were normalized to their absolute maximum, and colors denote the import rate ranging from 0% to 100% maximum import. Metabolite groups of interest are marked by blue and red bars to the right (branched-chain amino acids [BCAA]). Column headers are colored by the metabolic health state of the host (CTRL, control [metabolically healthy]; T1D, type 1 diabetes; T2D, type 2 diabetes; metformin+, with metformin treatment; metformin−, without metformin treatment). (B) Growth niche map for gut genera. Import fluxes for each genus in each sample were reduced to two dimensions using t-SNE. Each symbol denotes a genus in one sample and is colored and named by its genus. Genera that are close to each other consume similar sets of metabolites.

samples, where individual genera could be uniquely identified by their particular set of import fluxes (Fig. 4B). Here, taxa closer to one another overlap more in consumed metabolites (Fig. 4B). For instance, *Bacteroides* and *Prevotella* were relatively close to each other in the center of the map, which may help explain the observed trade-off between *Bacteroides* and *Prevotella* abundances in humans (45). *Blautia*, *Desulfovibrio*, *Bifidobacterium*, and *Eubacterium* had the most unique growth niches overall. Genus identity alone explained 61% of the variance in import fluxes (Euclidean permutational multivariate analysis of variance [PERMANOVA] $P = 0.001$). Thus, there was extensive growth niche partitioning between bacterial genera.

**SCFA production is driven by extensive cross-feeding within the microbiota and can be modulated by personalized interventions.** Given the association between SCFAs and disease phenotypes, we investigated the degree of SCFA production by the model microbiota (2, 46, 47). Intestinal cells have access to the full pool of SCFAs in the gut lumen and would probably take up a significant fraction of those extracellular SCFAs. Thus, the total export flux of any SCFA into the gut lumen by all taxa in a specific model is a measure of host-available SCFAs produced by the microbiota (see Materials and Methods for details on computation). Overall SCFA production for the major SCFAs showed large variations even in healthy individuals, which indicates a large impact of gut microbiota composition on SCFA availability. In particular, we observed that Swedish individuals showed higher SCFA production rates than the Danish individuals in the study. Butyrate production was diminished by about twofold in Danish individuals with type 1 diabetes (Welch's $t$ test, $P = 0.004$) but not in Swedish individuals. Danish and Swedish individuals with type 1 diabetes had microbiota that produced more acetate than healthy individuals (Welch's $t$ test, $P = 0.02$ and 0.003, respectively; Fig. 5A). Metformin treatment had a moderate effect in increasing butyrate production in both cohorts; however, this effect was strongest when comparing Danish individuals with type 1 diabetes (T1D) and metformin-treated Danish individuals with type 2 diabetes (T2D) (Welch's $t$ test, $P = 0.003$). Higher production of SCFAs was usually accompanied by an increased consumption of SCFAs within the gut microbiota (Fig. S5). This is consistent with prior findings in Danish and Chinese populations (4, 27, 48).

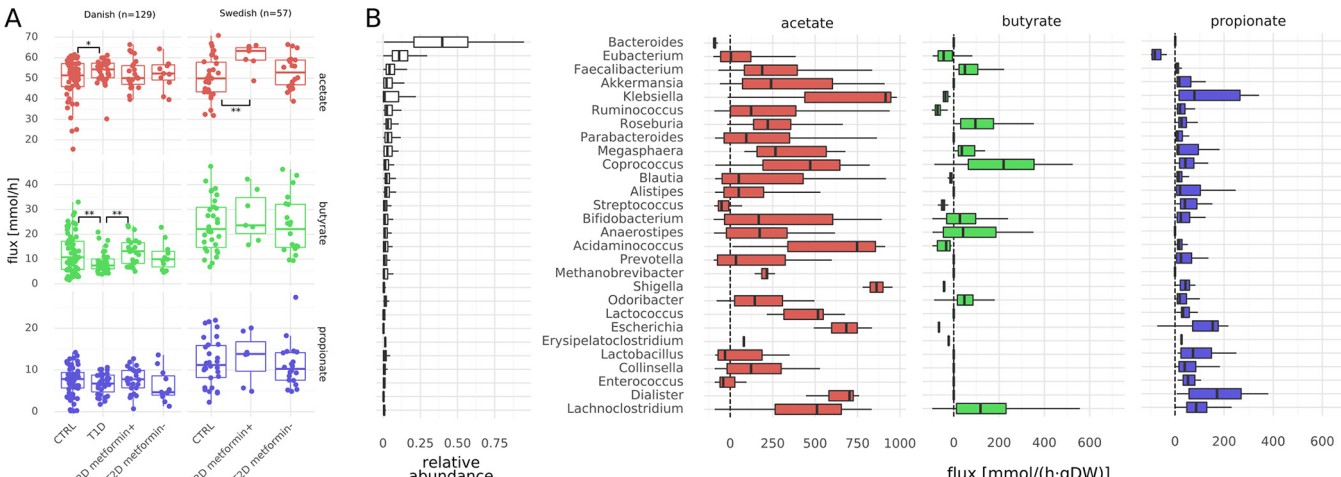

**FIG 5** SCFA fluxes. (A) Production capacities of the major SCFAs stratified by population. Fluxes denote total amount of SCFA produced by 1 g of bacterial biomass in the gut. Values that were significantly different by Welch's $t$ test are indicated by bars and asterisks as follows: *, $P < 0.05$; **, $P < 0.01$. (B) Genus-specific fluxes for the three major SCFAs. Only genera with a relative abundance of >1% are shown. Fluxes denote total production/consumption for each genus (see Materials and Methods) and are directed toward exports. Flux is shown in millimoles per hour per gram (dry weight). Thus, positive fluxes denote production of the metabolite, and negative fluxes denote consumption. Genera are ordered by average relative abundance (relative abundances are shown in the first column) from top to bottom.

Decreases in butyrate production were usually accompanied by increases in acetate production. This appeared to indicate SCFA cross-feeding within the microbiota, which we confirmed by comparing the total production and consumption fluxes for each bacterial genus for all samples (Fig. 5B). We observed that butyrate was almost exclusively formed in an acetate-dependent manner from acetyl coenzyme A (acetyl-CoA), which is the most prevalent butyrate production pathway in bacteria (49). In particular, production of butyrate depended on acetate production in the community. This was enabled by an extensive cross-feeding between the genera. All SCFAs were produced by a heterogeneous set of taxa, with acetate and propionate production being spread out for most taxa in the system and butyrate production being somewhat more restricted to a smaller set of taxa (Fig. 5B). Across samples, the most efficient butyrate producers were *Faecalibacterium*, *Coprococcus*, *Roseburia*, *Anaerostipes*, and *Lachnoclostridium*, all of which are known butyrate producers (49, 50). However, the models also predicted consumption of acetate by *Bacteroides* and consumption of butyrate by *Eubacterium*, which has not been commonly reported *in vivo*. Production of SCFAs was complemented by several other genera, generating a network of SCFA cycling within the microbiota. SCFA production by any genus showed high variation across samples and in some cases would even switch between consumption and production of a particular SCFA, which shows how specific SCFA production is to a particular microbial community (compare colored box plots in Fig. 5B). Net production of SCFAs was low compared to overall production (Fig. 5A and Fig. S5B), which indicates that most SCFAs in our models were cycled within the bacterial community.

Finally, as a proof of concept for the utility of MICOM, we aimed to quantify the impact of targeted interventions on the net consumption or production of SCFAs by the microbiota. To do this, we chose three Swedish samples from three individuals (healthy, T2D without metformin treatment, T2D with metformin treatment). The impacts of particular univariate interventions were then quantified by using elasticity coefficients (51, 52), which are dimensionless measures of how strongly a parameter affects a given flux (see Materials and Methods). Univariate interventions included increasing the availability of a single metabolite in the diet or increasing the abundance of a single bacterial genus. We observed that the effects of these single interventions were very heterogeneous for all three samples (Fig. 6). The strongest and most commonly observed effects were to diminish overall SCFA production. However, we

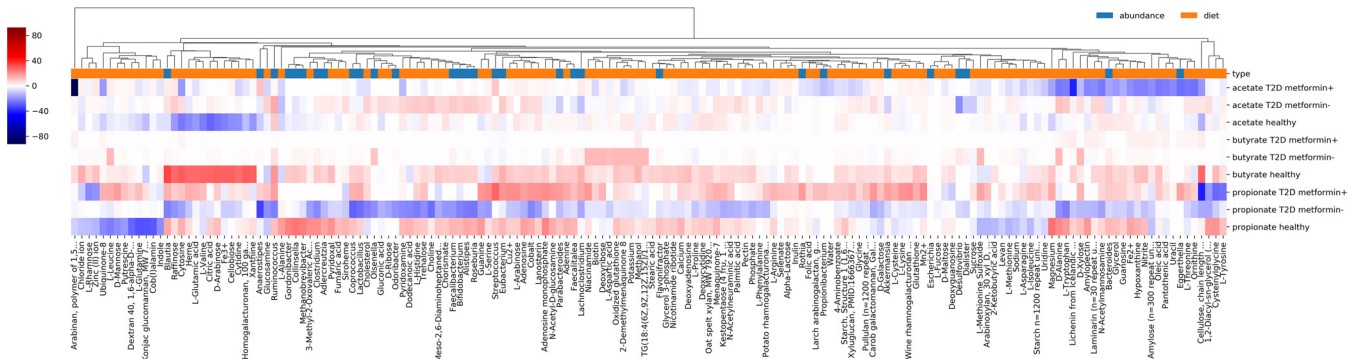

**FIG 6** Effects of interventions on SCFA production in three samples. Each row denotes an SCFA in a specific individual, and each column denotes either a diet component or bacterial genus. Colors denote the elasticity (i.e., the percent change in SCFA production given a percent increase in the specific effector). Red denotes interventions that would increase SCFA production, and blue denotes interventions that would decrease production. Only interventions with nonzero elasticities in at least one sample are shown.

observed a few interventions that were able to weakly increase SCFA production. There was a distinct set of metabolites that would increase butyrate production in the T2D individuals but not in the healthy individual. This was also dependent on metformin status. For instance, arabinan increased butyrate production in the metformin-treated individual, and D-xylose increased butyrate production in the non-metformin-treated individual, but not *vice versa*. Thus, MICOM is able to explore the potential functional consequences of targeted dietary or probiotic interventions, which can differ greatly depending on the context of the microbiota in which the interventions are made.

## DISCUSSION

There is a large amount of sequencing data on microbial communities available today. This is due in part to the falling cost of 16S rRNA and shallow shotgun sequencing (53). There is wide interest in extracting information from sequencing data that goes beyond bacterial proportions (54). Metabolic modeling incorporates a rich knowledge base from genomics and biochemistry and is a valuable resource for adding value to existing sequencing data sets. Specifically, MICOM allows for the integration of genome-scale metabolic models, dietary information in the form of import flux bounds, and abundance estimates from metagenomic shotgun or marker gene sequencing. This framework enables *in silico* predictions concerning ecological interactions within microbial communities, inferred exchanges between microbial communities and their environment, and mechanistic hypotheses for how metabolic interactions can be modulated by changes in the environment. The design of reasonable metagenome-scale metabolic models is challenging due to apparent trade-offs between individual and community growth rates and issues with computational tractability. Here, we provided a viable strategy that allows for complex analysis of the metabolic consequences of variation in microbial community composition. Our regularization strategy allows for fast identification of unique sets of individual growth rates, which operate in biologically realistic ranges. Our assumption that there is a trade-off between community growth rate and individual taxon growth rates is supported by the observation that most microbial communities are composed of a large number of species with nonnegligible abundances. Individual growth rates for bacterial genera varied greatly across samples (Fig. 2) and were tightly coupled to genus abundances within a sample (Fig. 3A). However, there may be other regularization strategies that provide better agreement with the underlying biology. Our validation strategy using replication rates obtained directly from metagenomic data provides a simple framework to test new regularization functions in the future. It seems that the large variation of growth rates can be explained by the dependence of the growth rate on the presence of other bacteria in the sample (Fig. 3B). Thus, bacterial growth in the gut microbiota is dictated not only by abundance but also by taxon-taxon interactions.

Our predictions are somewhat limited by a variety of factors. For instance, the lack of metabolic models for the major cell types of the gut epithelium (especially goblet cells, enterocytes, and colonocytes) and sample-specific metabolite availability in the gut lumen limits the accuracy of MICOM's predictions. Additionally, the use of representative models for bacterial genera is a notable caveat. Available metabolic reconstructions are often based on laboratory strains that may not represent the exact metabolic capacity of strains in the human gut. Thus, reconstructions may lack certain metabolic pathways present in the sample and yield inaccurate results, especially when not applying appropriate bounds for the underlying diet (33, 55). Thus, the inferred metabolic exchanges should be seen as qualitative and will require validation against quantitative data. Model predictions may become more quantitative as better (more personalized) data become available. The incorporation of personalized data on diet and a better grasp of what fraction of each metabolite is absorbed in the small intestine should help to improve model-based predictions. Personalized reconstruction of microbial community metabolic models directly from metagenomic sequencing data may provide more accurate predictions as well, but this approach is currently limited by insufficient sequencing coverage for low-abundance taxa.

MICOM provided valuable ecological insights into the gut microbiota. For instance, the cooperative trade-off in equation 2 indicates that a more diverse microbiome (i.e., higher evenness) results in higher individual growth rates (on average) due to the magnitude scaling of the abundance vector (denominator in equation 2). We also found strong niche partitioning in the model, where taxa showed minimal overlap with each other in resource utilization space. This minimal overlap implies that there is likely an upper bound on alpha diversity in the gut, due to the fact that growth niches eventually saturate and limit the number of taxa that can engraft. Even though only about a third of metabolites were co-consumed by any pair of taxa in the models, this small amount of niche overlap still resulted in resource competition between taxa. This was particularly true for dominant taxa (e.g., *Bacteroides*), which tended to show competitive interactions with many other genera, likely due to the comparatively higher resource requirements of these abundant taxa for maintaining growth. This community-wide resource competition fits well with the observed growth dependence on amino acid import fluxes for all taxa (Fig. 3A), which is consistent with prior work that suggest that nitrogen may be the global limiting factor for microbial growth in the gut (56). Finally, the methods here extend to any ecosystem containing many bacterial taxa. As such, MICOM can be employed to perform functional analyses on a wide range of microbial ecosystems.

It has been difficult and time-consuming to obtain empirical evidence for the mechanistic basis of gut-microbiota interactions. MICOM provides a high-throughput platform for generating mechanistic hypotheses and running *in silico* experiments that would be impossible to perform *in vivo*. Thus, we feel that the major application for MICOM is to provide detailed functional hypotheses that can serve as targets for experimental validation. For example, MICOM reveals widespread SCFA cross-feeding in the gut microbiota. The mere presence of butyrate producers was not enough for stable butyrate production—acetate production was also required. Furthermore, MICOM generated personalized predictions for how dietary and probiotic interventions influenced SCFA production capacity. This basic approach could be extended to any number of clinically relevant metabolites. Thus, we hope that the method presented here will aid researchers in leveraging existing gut microbiome data to design and test personalized intervention strategies.

## MATERIALS AND METHODS

**Metagenomic shotgun data analysis.** All metagenomic analyses were performed in R using an in-house pipeline which is available as an open-source package along with documentation at https://github.com/resendislab/mbtools. Sample FASTQ files were downloaded using the SRA toolkit and trimmed and filtered using the DADA2 "filter_and_trim" function (57) with a left trimming of 10 bp, no right trimming, a quality cutoff of 10, and a maximum number of two expected errors under the Illumina model. Abundances for different taxa levels were then obtained using SLIMM (29) which was chosen

mSystems®

because it supported one of the largest references (almost 5,000 reference bacterial genomes). In brief, all sample FASTQ files were first aligned to the SLIMM reference using Bowtie2, saving the 100 best matches for each read. Taxon abundance profiles were then obtained using SLIMM with a window size of 100 bp and assembled into a single abundance file. SLIMM coverage profiles resolved to single strains were then used to infer replication rates using the iRep method (58). In brief, coverage profiles were first smoothed with a rolling mean over 5-kbp windows, and only genomes with at least a mean coverage of 2 and with at least 60% of total length covered were considered. Coverage values were log transformed and sorted, and the lowest and highest 10% of the data points were removed to obtain the linear part of the curve. Replication rates were then inferred from the slope of a regression on the linear portion. An estimate for the minimum coverage was then obtained from the intercept of the regression, and only replication rates for strains with a minimum coverage of >2 were kept. No correction for GC content was performed. Before model construction, genus-level and species-level quantifications for each sample were matched separately to the AGORA models (version 1.03) by name. The final quantification and mapping are provided in the data repository ("genera.csv" and "species.csv" at https://github .com/micom-dev/paper).

The growth rates predicted by MICOM were then compared to the mean replication rate for each taxonomic rank (either species or genus) in each sample. Concordance was quantified by Pearson correlation either within each sample with at least six measured replication rates or for all samples. Dependence on outliers was checked by an inspection of log-log plots of predicted growth rates versus measured replication as well as by running the same analysis using Spearman correlations, which yielded similar results.

**Strategies used in MICOM.** MICOM is based on the popular COBRApy Python package for constraint-based modeling of biological networks and is compatible with its application programming interface (API) (59). The cooperative trade-off strategy as described here was introduced to MICOM in version 0.9.0. Flux balance analysis obtains approximate fluxes for a given organism by assuming a steady state for all fluxes in the biological system and optimizing an organism-specific biomass reaction. Using the stoichiometric matrix $S$ which contains reactions in its columns and metabolites in its rows, this can be formulated as a constrained linear programming problem for the fluxes $v_i$ (in millimoles per gram [dry weight {DW}] per hour {mmol/[gDW h]}):

$$\text{maximize } v_{bm}$$

$$\text{such that (s.t.) } Sv = 0$$

$$lb_i \leq v_i \leq ub_i$$

The biomass reaction $v_{bm}$ is usually normalized such that it will produce 1 g of biomass which results in a unit 1/h corresponding to the growth rate $\mu$ of the organism. The upper and lower bounds ($ub_i$ and $lb_i$, respectively) impose additional thermodynamic constraints on the fluxes or restrict exchanges with the environment (in the case of exchange fluxes). In order to describe a community model containing several organisms with each organism having a particular abundance $a_i$ (in grams [DW]), one usually embeds each organism in an external compartment which represents the community environment (for instance, the gut lumen for models of the gut microbiota). Adding exchanges for the environment compartment and exchanges between a particular organism and the environment, one obtains a community model with the following constraints:

$$\text{maximize } \mu_c = \sum_i a_i \mu_i$$

$$\text{s.t. } \forall\, i : Sv = 0$$

$$\mu_i = v_i^{bm} \geq \mu_i^{min}$$

$$lb_i \leq v_i \leq ub_i$$

$$lb_i^{ex} \leq a_i v_i^{ex} \leq ub_i^{ex}$$

$$lb_i^m \leq v_i^m \leq ub_i^m$$

Here, $a_i$ denotes the relative abundance of genus $i$, $\mu_i$ is its growth rate, $v_i^{bm}$ is its biomass flux, $\mu_i^{min}$ is a user-specified minimum growth rate, $v_i^{ex}$ represents the exchange fluxes with the external environment, and $lb$ and $ub$ are the respective lower and upper bounds. Additionally, $\mu_c$ denotes the community growth rate and $v_i^m$ represents the exchanges between the entire community and the gut lumen.

The described constraints are identical to the ones employed in SteadyCom (22, 29). However, SteadyCom aims to predict microbial abundances from a list of taxa present in a sample, whereas MICOM predicts growth rates and fluxes and requires abundances as input. In particular, SteadyCom assumes that all taxa grow at the same rate as they are all subject to the same dilution rate in the SteadyCom model. Previous studies have shown that bacterial growth rates in the human gut vary widely across taxa (30). Thus, MICOM allows for taxon-specific dilution and growth rates.

We assigned an upper bound of 100 mmol/[gDW h] for the internal exchange fluxes $v_i^{ex}$. Assuming a total microbiota biomass of 200 g and a representative bacterial cell weight (dry weight) of 2 pg (60), this corresponds to a maximum import or export of more than 100,000 molecules per cell per s. Diet-derived lower bounds with values smaller than $10^{-6}$ mmol/[gDW h] were set at zero, as they would have been lower than the numerical tolerance of the solver. Taxa with relative abundances $a_i$ smaller than $10^{-3}$ for the genus models or $10^{-4}$ for the species models were discarded, since they would not be able to affect the external metabolite levels in a significant way but do increase computation time.

Internal fluxes $v_i$ received respective bounds of 1,000.0 (or 0 if irreversible), making them essentially unbounded. The described constraints are applied to all optimization problems in MICOM and will be further called the "community constraints."

The cooperative trade-off method consists of two sequential problems. First, maximize the community growth rate $\mu_c$ to obtain $\mu_c^{max}$. Using a user-specified trade-off $\alpha$, now solve the following quadratic minimization problem:

$$\text{minimize} \sum_i \mu_i^2$$

$$\text{s.t. } \mu_c \geq \alpha \mu_c^{max}$$

and community constraints

The knockout for a genus $i$ was performed by setting all fluxes belonging to this genus along with its exchanges with the external environment to zero (lb = 0 and ub = 0). This is followed by running cooperative trade-off on the knockout model and comparing the growth rates after the knockout with the ones without the knockout.

**Solvers and numerical stabilization.** Most genome-scale metabolic models usually do not treat more than 10,000 variables in the corresponding linear or quadratic programming problems. However, in microbial community models, we usually treat tens to hundreds of distinct genome-scale models, which makes the corresponding problem much larger. Unfortunately, many open-source and commercial solvers have difficulties solving problems of that scale, so we also implemented strategies to increase the success rate of those optimizations. All linear and quadratic programming problems were solved using interior point methods, as these methods were much faster than simplex methods for problems with more than 100,000 variables. Here, we used CPLEX (https://cplex.org) but also tested all methods with Gurobi (https://www.gurobi.com/) (see "Data availability" below). Since growth rates tend to be small, we also multiplied the objectives used in the cooperative trade-off (maximization of community growth rate and minimization of regularization term) with a scaling factor in order to avoid near-zero objective coefficients. A scaling factor on the order of the largest constraint (1,000.0) seemed to work well. Nevertheless, the default interior point methods for quadratic problems in CPLEX or Gurobi were usually not capable of solving the minimization of the regularization term to optimality and usually failed due to numerical instability. The solutions reported by the aborted optimization run were usually close to the optimum but tended to violate some numerically ill-conditioned constraints. To alleviate this problem, we implemented a crossover strategy where we took the solution of the numerically ill-conditioned quadratic interior point method as a candidate solution set $\mu_i^{ca}$. On the basis of that, we then optimized the following linear programming problem in order to restore feasibility:

$$\text{maximize } \mu_c = \sum_i a_i \mu_i$$

$$\text{s.t. } \mu_i \leq \mu_i^{ca}$$

and community constraints

Linear interior point methods are usually numerically stable so this linear programming problem can usually be solved to optimality. The maximization together with the new constraints will push the individual growth rates toward the candidate solution as long as it is numerically feasible.

**Minimal medium and exchange fluxes.** By convention, MICOM formulates all exchange fluxes in the export direction so that all import fluxes are positive and export fluxes are negative. Based on this, the minimal medium flux for a community was obtained by minimizing the total import flux:

$$\text{minimize } v_{tot} = \sum_i \left\{ |v_i^m|, \ v_i^m < 0 \right\}$$

$$\text{s.t. } \forall \, i : \mu_i \geq \mu_i^{ct}$$

$$\mu_c \geq \alpha \mu_c^{max}$$

and community constraints

Here, $\mu_i^{ct}$ denotes the optimal genus growth rates obtained by cooperative trade-off. The community exchanges were then obtained by extracting all $v_i^m$, whereas genus-specific exchanges were given by all $v_i^{ex}$ as defined earlier.

Overall production fluxes were calculated as

$$v_{tot}{}^m = \sum_{i,v_i^m > 0} a_i v_i^m$$

where $v_i^m$ denotes an exchange flux for the metabolite $m$ in taxon $i$. Overall consumption rates were calculated in a similar manner but restricting fluxes to ones with $v_i^m < 0$ (imports).

**Single target intervention studies.** We used elasticity coefficients (51, 52) to evaluate the sensitivity of exchange fluxes to changes in exchange flux bounds (*ergo* diet changes) or changes in genus abundances. The logarithmic formulation of elasticity coefficients is given by

$$\varepsilon_p{}^v = \frac{\partial \ln|v|}{\partial \ln|p|}$$

where $v$ denotes the exchange flux of interest and $p$ is the changed parameter. Since the absolute value removes information about the directionality of the flux, this was logged separately to maintain this information. We used a value of 0.1 as differentiation step size in log space, which corresponds to a

bound or abundance increase of about 10.5% in the native scale. To enable efficient computation, elasticity coefficients were grouped by the *p* parameter, then the cooperative trade-off was run once without modification, the *p* parameter was increased, the cooperative trade-off was run again, and differentiation was performed for all exchange fluxes at once.

**Data availability.** All data to reproduce the manuscript, intermediate results as well as Python scripts to reproduce the figures in the manuscript are available in a data repository at https://github.com/micom-dev/paper. Metagenomic reads for the 186 individuals were obtained from a balanced sample set generated from the SRA BioProjects PRJEB4336, PRJEB5224, and PRJEB1786 as published by Forslund et al. (27) and can be downloaded from the Sequence Read Archive (https://www.ncbi.nlm.nih.gov/sra) with the SRA toolkit (https://github.com/ncbi/sra-tools). An exhaustive list of accession identifiers (IDs) for the individual samples is provided in the data repository (https://github.com/micom-dev/paper/blob/master/data/recent.csv). All algorithms and methods used here were implemented in a Python package and can be easily applied to different data sets. The Python package MICOM (for *mi*crobial *com*munities) along with documentation and installation instructions are available at https://github.com/micom-dev/micom. A QIIME2 plugin for MICOM (q2-micom) is available at https://github.com/micom-dev/q2-micom. The AGORA reference reconstructions (version 1.03) can be downloaded from https://vmh.uni.lu/#downloadview. The Western average diet fluxes along with used dilution factors have been deposited at https://github.com/micom-dev/paper/blob/master/data/western_diet.csv. Several methods used in MICOM require an interior point solver with capabilities for quadratic programming problems (QPs) for which there is currently only commercial software available. MICOM supports CPLEX (https://cplex.org) and Gurobi (https://www.gurobi.com/) both of which have free licenses for academic use. Intermediate results that required those solvers are also provided in the data repository to permit reproduction of our major conclusions.

## SUPPLEMENTAL MATERIAL

Supplemental material is available online only.

**TEXT S1**, PDF file, 0.1 MB.
**FIG S1**, TIF file, 1.2 MB.
**FIG S2**, TIF file, 0.4 MB.
**FIG S3**, TIF file, 0.6 MB.
**FIG S4**, TIF file, 0.5 MB.
**FIG S5**, TIF file, 0.6 MB.

## ACKNOWLEDGMENTS

O.R.-A. and C.D. were supported by an internal grant from the National Institute of Genomic Medicine (INMEGEN/México). S.M.G. and C.D. were supported by a Washington Research Foundation Distinguished Investigator Award and startup funds from the Institute for Systems Biology.

C.D. developed/implemented the methods and performed the analysis. S.M.G. helped design the metagenomic analyses and growth rate validations. O.R.-A. developed the methods and designed the meta-analysis. All authors wrote the manuscript.

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
