## [Reviewer comments · mSystems]

MICOM: metagenome-scale modeling to infer metabolic interactions in the gut microbiota

Christian Diener, Sean Gibbons, and Osbaldo Resendis-Antonio

Corresponding Author(s): Osbaldo Resendis-Antonio, Coordinacion de la Investigacion Cientifica - RAI UNAM

Review Timeline:

Submission Date:	September 20, 2019
Editorial Decision:	November 4, 2019
Revision Received:	December 16, 2019
Accepted:	December 19, 2019

Editor: Nicholas Chia

Reviewer(s): The reviewers have opted to remain anonymous.

Transaction Report:

DOI: <https://doi.org/10.1128/mSystems.00606-19>

Response to reviewers

All mentioned line numbers refer to the **marked-up manuscript**.

Reviewer #1 (Comments for the Author)

In this manuscript, the authors suggest that they built the community metabolic model of the human gut microbiota, and analyzed the metagenome data based on this model. Although the motivation of this study is great, I raise several serious concerns about the validity, rationale, and validation of their models, and presentation of their methods and data.

We thank the reviewer for their valuable comments. We have substantially revised the text and added additional analyses and figures to address reviewer concerns. In particular, we performed a validation analysis using >1000 replication rate measurements obtained from the metagenomic sequencing data and show good correspondence between MICOM growth rate predictions to observed replication rates. See new text at lines 279-289 and Figures 1C and S1.

-- To my best understanding, this work does not present real mechanistic model, because there is no demonstration that this model has been used for de novo calculation (or prediction) of the time-course population of each microbial species. This paper seems to use only a priori known abundance of each microbial species (from metagenome data) and only calculates "growth rates" of these species, and no use of these growth rates for the calculation of dynamic population changes over time. Such "static" scheme may merely reflect the limitation of the scope of this study, rather than the critical defect of this study; however, I wonder how such static scheme actually predicts the nature of metabolic interactions between microbes (which is major claim of this paper). In fact, whether the metabolic interactions between two microbial species is competitive or cooperative, it will be dependent on the specific concentrations of metabolites inside the medium, as suggested in many previous FBA studies. With such a static scheme in this paper, it is not mathematically feasible to calculate such concentration of metabolites, and therefore it is hard to accept the main claim of this paper on the nature of metabolic interactions between microbial species.

We agree with the reviewer that FBA is not a dynamic model of bacterial growth and may not capture dynamic changes of population structure. Capturing dynamics would require much more detailed knowledge of individual kinetics, including several thousand to hundreds of thousands of kinetic parameters. FBA usually assumes that growth rates are in steady state, which is biologically realistic during log-phase growth in a chemostat-like environment. Similar to a chemostat, real-world microbial communities are often balancing growth with some constant rate of dilution (i.e. either by cell death or physical removal from the system). Under these

conditions, growth rates need to be balanced with dilution, which would result in approximately constant abundances over time. Recent work has shown that, absent major perturbations or persistent changes in diet, the average within-person abundance of any given gut bacterial species does not change over months-to-years (<https://doi.org/10.1371/journal.pcbi.1005364> ; <https://doi.org/10.1038/s41591-019-0559-3>). In other words, gut bacterial species appear to maintain stable carrying capacities within a person. This is consistent with a steady state community model as previously derived by Hung et. al. (<https://doi.org/10.1371/journal.pcbi.1005539>). We have included additional references and text to address these points in the Results section (lines 147-156).

-- Also, this paper claims that they used Western diet for their simulation. However, it is difficult to find in the manuscript their specific method of how they connected Western diet to actual nutrient content in the human large intestine. Inferring this connection is highly nontrivial task, because different nutrients in dietary foods have different digestion and absorption rates in upper GI tract.

We agree that this was not described in sufficient detail. As the reviewer suggests, our initial approach did not account for the uptake of metabolites by the host in the small intestine. Unfortunately it is difficult to find exact uptake rates for each of the modeled metabolites in the small intestine. To circumvent this we depleted all classes of metabolites known to be absorbed in the upper GI tract by tenfold and provide a better description for how the particular diet was applied to the models (lines 310-317, 608-621).

-- In addition, the quality of their figures makes it hard to evaluate whether the claims in this manuscript are actually supported by the data in their figures.

We have updated all figures in the manuscript and restructured the main text to better support the claims made in the paper. We welcome any suggestions from the reviewer to further improve the clarity of our work.

-- Fundamentally, this work lacks the solid mathematical/biological ground of their model formulation. Eq (1) describes community growth rates that they want to optimize. However, such optimization at the community level has long been questioned regarding its relevance to the real-world micro-ecology, although individual organism-level optimization can be accepted as the estimation of that microbial growth (as in usual FBA studies so far). The authors seem to be aware of this issue, and therefore they tried to use the fraction of the maximum of the community growth rates. However, even in this case, it is not yet clear how this specific optimization scheme of Eq. (1) can be rigorously justified. For example, Eq. (1) means the "average" growth rates of microbes. If someone writes ODE of population growth dynamics of the microbial community, it becomes immediately obvious that using of such "average" growth rates as the community-representative quantity is only valid for very short-time periods (shorter than or comparable to doubling times of the fastest-growing microbes), and therefore using of Eq (1) itself sounds rather superficial. Also, about their regularization to select unique solution

among multiple solutions, note that regularization is commonly used to select "sparse" solutions among multiple solutions. However, this study used L2 regularization to avoid the exactly such "sparse" solution, i.e. they excluded L1 regularization because it didn't give the even distribution of growth rates of different microbial organisms. If this work prefers L2 regularization that gives the even distribution, I wonder why this work considered regularization itself among many alternative methods of selecting unique solutions. Without rigorous justification of this point, it will be difficult to understand what is the mathematical rationale of the use of regularization itself in this study. About this point, this study mentioned that the use of such regularization enables the growth of many species as observed in experimental data. However, only relying on such "empirical" observation cannot constitute the solid ground of the use of particular mathematical formulas, because the result of such formulas (without mathematical rigor) are likely to be the methodological artifacts made by the fitting, although it may show some qualitative agreement with empirical observation.

Those are all excellent points, which brings us back to a discussion about dynamics. As the reviewer points out, the model growth rates can be thought of as time-integrated averages. However this is also true for sequencing data taken from fecal samples. A single stool sample cannot be used to capture dynamics. Furthermore, even if a time series of stool samples were collected (i.e. a sample collected once per day), the collection time is much coarser than gut bacterial growth rates (i.e. hours; <https://science.sciencemag.org/content/349/6252/1101>). Thus, stool samples represent the endpoint of dynamics and reflect the approximate steady-state composition of the gut microbiome (<https://doi.org/10.1371/journal.pcbi.1005364>), which makes these data adequate for the proposed modeling framework. We have added some discussion of these points to the text (lines 147-159).

Our motivation to use L2 regularization was indeed not well justified. We now justify the regularization from the intuitive principle that taxa observed at appreciable abundances in the data should be able to grow (Supplemental Text S1). We show that no linear regularization scheme can fulfill that principle but that L2 regularization does. In particular, we derive an approximate closed form solution for the growth rates given the L2 regularization (Supplementary Text S1). Most importantly, we show that using L2 regularization strongly increases the correlation with observed replication rates, whereas a lack of regularization leads to growth rate predictions that are totally uncorrelated with observed replication rates (lines 319-353, Fig. 1C, S1 and S2). Finally, we provide better discussion of alternative regularization schemes in the text (lines 596-603).

-- Also, this work used AGORA models. However, it is known that the level of manual curation of AGORA models is not that comprehensive, and these models show very poor predictability of the experimental data [Nat Microbiol. 2018 Apr;3(4):514]. For example, one can easily check that some of AGORA models can even secrete oxygen gas. Without the rigorous quality assessment and extensive curation of their models, it is difficult to justify the validity and accuracy of the models in this study.

The reviewer raises an important point. AGORA models are definitely not perfect but show good performance as long as a realistic set of dietary import fluxes is assumed. As the original authors show in their reply to the cited correspondence piece, applying the Western diet (used in the current study) yields better growth rate estimates and prevents the issues raised in the correspondence cited by the reviewer (<https://doi.org/10.1038/nbt.4212>). We now cite and mention those drawbacks and suggest alternatives in the discussion (lines 611-621). Most importantly, our extensive validation now shows that even though the AGORA model may be imperfect, it still yields biologically reasonable results. For example, our model results clearly show anaerobic growth of communities across all samples. We do not expect any mechanistic model of a complex microbial community to be completely accurate, and the extent to which the model breaks down or provides non-intuitive results helps us to improve our understanding of the underlying biology.

-- About the validation of the model in this study: This paper claims that the overall growth rate of their models match with experimental data, using the data in Ref. [27]. However, the growth rates in Ref. [27] is based on pure computational inference (based on mutational rates on genome), and therefore too speculative data to serve as a validation data of their model. Also, Table 1 in Ref. [27] only shows a few microbes which are too phylogenetically-restricted; therefore, it is rather suspicious whether Table 1 in Ref. [27] can really provides overall growth rates of the microbes inside a human gut. The authors claim that doubling time of ~20 min in their model capture the fastest growth rate of micrbes in laboratory. Actually, this growth rate corresponds to E. coli's fastest growth rate in laboratory. However, to my knowledge, there is no experimental report that E. coli can also achieve such a level of growth rate in the gut environment, and it is very unlikely for E. coli to achieve such growth rate regarding the anaerobic gut environment. I suggest that, for the validation of the growth rates of their models, the authors need to use the more realistic growth rate data relevant to the in vivo gut environment, such as those in Science 2015 Sep 4;349(6252):1101.

We thank the author for the excellent suggestion, which has greatly improved the quality of our manuscript. We now use the iRep method (<https://doi.org/10.1038/nbt.3704>), which is based on the reviewer-cited paper from Eran Segal's group, as it yields replication rate estimates for a larger number of taxa than the original method and does not require exact localization of the origin of replication. We show that regularization greatly improves the correspondence between replication rates and the model growth rates. In the absence of regularization, there is essentially no correlation between the observed replication rates and model growth rates (lines 319-353).

-- About the statistical result in this paper: L245 - L247 suggests the power-law relation of their result, but this argument sounds superficial and does not make much sense. For the criterion of power-law statistics, please refer to Aaron Clauset or Mark Newman's review papers.

We agree with the reviewer and have now removed that problematic figure and section of the manuscript. Instead, we have derived a closed form approximation for replication rates that

explains the correlation between abundances and growth rates (main text lines 219-260, 379-394 and Supplementary Text S1).

-- The citation of each reference should be thoroughly checked whether cited correctly. For example, I see that the citation of Ref 27 misses either its volume number or page number.

We have corrected the mentioned citation and have made an effort to ensure that all citations are accurate and complete.

-- The writing style and English should be thoroughly improved throughout the manuscript. The examples of such phrases include "one being just reporting any one of the possible growth rates distributions" and "is composed by". Actually, some sentences are difficult to interpret their meanings, obscuring the detailed ideas and methodologies taken in this manuscript.

The manuscript has been thoroughly revised and rewritten in several sections and has now been checked by multiple native English speakers.

Reviewer #2 (Comments for the Author):

Major comments:

1) *In my opinion it is hard to assess the validity of the central idea of the micom method (the specific way to compute tradeoff between overall community and individual species growth) without some validation using a smaller model community where some of the nutrient exchanges are known. The method should presumably work for small communities just as well and a more controlled setup would allow for example determining the validity of the minimal nutrient import hypothesis.*

We thank the reviewer for raising these concerns. We have now included an extensive validation of our method using a set of >1,000 *in vivo* replication rates in order to show that the regularization performed by MICOM greatly improves correlation with the observed replication rates (Fig. 1C, S1 and S2, lines 319-353). In particular metabolic modeling without a regularization strategy shows essentially no correlation with observed replication rates. Nutrient imports have been bounded based on a previously reported average Western diet, which is now described in more detail in the text (lines 310-317). Furthermore we now discuss the potential pitfalls and agreements of the observed minimal import fluxes in the main text and discussion (lines 608-640).

2) *Due to the lack of availability of species-level genome-scale models for approximately 1/3 of the species that are observed in the human gut microbiome, the authors resort to using genus level models. It is not clear to me how much functional information is lost in this*

transformation. For example, in cases where species level models are available within a specific genus, would including the species-level models change the predictions significantly? This would be expected to be the case if there are distinct species within a genus that have significantly different metabolic behaviors and also vary significantly in abundance between different individuals.

We now included a comparison of genus vs. species level modeling of the microbial communities (lines 293-305, 348-353, Fig. S2). We observe that species and genus level models both can give good agreement with *in vivo* replication rates (Fig. 1 and Fig. S2). We do observe some discrepancies for genus level models when not applying a growth tradeoff (enforcing optimal or near-optimal community growth) where genus level models can show negative correlations with *in vivo* replication rates. This may very well be due to unrealistic metabolic functions within the genus models. However, genus level models did agree well with *in vivo* data when applying the growth tradeoff, indicating that species and genus level models both perform well in that particular setting. Furthermore, most available data sets (in particular, 16S amplicon sequencing) do not allow for species-level annotations. Thus, we feel that sticking with genus level models is appropriate here. The comparison between genus and species models is included in shown additionally in the new Supplemental Figure S2.

3) The manuscript is not very clear on how the nutrients available to the gut microbiome are determined. In my reading, these are determined solely through minimization of nutrient import fluxes that allow a reasonable trade off between growth of the overall community and individual species. It is not clear to me from the manuscript if the inferred nutrient imports are fully consistent with what is known about nutrient availability in human gut. The manuscript should more explicitly specify that no a priori nutrient composition information was included and provide better evaluation of the predictions made by the model.

We apologize for our lack of clarity in the previous draft of this manuscript. We actually did use established information based on the nutrient composition of an average Western diet. Additionally, we now deplete metabolites that are commonly taken up in the upper GI tract by 10-fold prior to running the models. This is now clarified in the text (lines 310-317). The detailed list of used flux bounds for that is now also clearly linked in the text (line 889). However, in the absence of sample-specific dietary data we assumed the same Western diet for all of the Danish and Swedish participants. This is now highlighted better in the text and in the discussion (lines 310-317, 616-618). Given the correlation between the predicted growth rates and observed replication rates, our approach appears to generate biologically realistic outputs. However, we hope that the bioinformatic research community will help us to improve MICOM's accuracy by identifying clever ways to integrate custom diets and by including more detailed metabolic models of host tissues.

Minor comments:

1) *Line 227: Is it possible to compare model predictions of viable species to what has been experimentally determined?*

Yes, this is now done using *in vivo* replication rates and appears in Fig. 1C, S1 and S2.

2) *Figure 1A: It's not clear what L1 and L2 actually denote here - they are not specified in the caption or the manuscript in general. I do understand that they refer to L1 and L2 norms, but the manuscript needs to contain equations defining these clearly.*

We have included more explicit definitions of L1 and L2 norms in the caption of Fig. 1.

3) *Figure 1B: It would be better to specify that growth/no growth cutoff $1e-6$ in the figure than in the caption.*

This is now done as a shaded area.

4) *Figure 1C: The meaning of the dashed line is not defined in the figure or its caption.*

Figure 1C is now replaced by the correlation figure and the meaning of the zero line in this figure has been added to the caption.

5) *Figure 2: The sorting criterion is not specified in the figure caption and whatever is used as a sorting criterion should also be plotted (e.g. median or mean growth rate).*

The sorting criterion (mean growth rate) is now mentioned in the caption and shown in the plot.

6) *Figure 4C: Show numbers of individual samples in each category in the plot or legend.*

This is now shown in the column headers of the new Figure 5A.

7) *Figure 6: Panels (A,B,C) not defined in the legend. Also, if the goal is just to show that the control sample shows higher "robustness" than the diabetic samples, it would be better to just show a simple plot that shows that number of times there is a significant flux response to a perturbation.*

The intention of this figure was to illustrate that one can predict interventions for SCFA production in a personalized manner. This panel has now been replaced by a new figure, which should be clearer and only shows the impact on the production fluxes of the major SCFAs (Fig. 6). The entire section on the elasticities has also been rewritten to be more specific and clear of this point (lines 563-578).

November 4, 2019

Dr. Osbaldo Resendis-Antonio
Coordinacion de la Investigacion Cientifica - RAI UNAM
Mexico City
Mexico

Re: mSystems00606-19 (MICOM: metagenome-scale modeling to infer metabolic interactions in the gut microbiota)

Dear Dr. Osbaldo Resendis-Antonio:

Below you will find the comments of the reviewers.

To submit your modified manuscript, log onto the eJP submission site at <https://msystems.msubmit.net/cgi-bin/main.plex>. If you cannot remember your password, click the "Can't remember your password?" link and follow the instructions on the screen. Go to Author Tasks and click the appropriate manuscript title to begin the resubmission process. The information that you entered when you first submitted the paper will be displayed. Please update the information as necessary. Provide (1) point-by-point responses to the issues raised by the reviewers as file type "Response to Reviewers," not in your cover letter, and (2) a PDF file that indicates the changes from the original submission (by highlighting or underlining the changes) as file type "Marked Up Manuscript - For Review Only."

Please return the manuscript within 60 days; if you cannot complete the modification within this time period, please contact me. If you do not wish to modify the manuscript and prefer to submit it to another journal, please notify me of your decision immediately so that the manuscript may be formally withdrawn from consideration by mSystems.

To avoid unnecessary delay in publication should your modified manuscript be accepted, it is important that all elements you upload meet the technical requirements for production. I strongly recommend that you check your digital images using the Rapid Inspector tool at <http://rapidinspector.cadmus.com/RapidInspector/zmw/>.

Sincerely,

Nicholas Chia

Editor, mSystems

Journals Department
Reviewer comments:

Reviewer #1 (Comments for the Author):

In this revision, the authors have addressed several issues raised by the reviewers' comments and I appreciate their efforts. However, I think that the revision is far from satisfactory and it detours the critical points in my comments. Also, the validation of the model is not yet clear, and the mathematical ground of their method, for example the regularization part in Supplementary Text, seems to involve mathematical flaws.

1. My previous comments include "however, I wonder how such static scheme actually predicts the nature of metabolic interactions between microbes (which is major claim of this paper). In fact, whether the metabolic interactions between two microbial species is competitive or cooperative, it will be dependent on the specific concentrations of metabolites inside the medium, as suggested in many previous FBA studies. With such a static scheme in this paper, it is not mathematically feasible to calculate such concentration of metabolites, and therefore it is hard to accept the main claim of this paper on the nature of metabolic interactions between microbial species."

==> In the authors' rebuttal letter, I do not think that the authors gave a satisfactory answer to this major comment. To my best interpretation, the authors gave only the justification of why FBA can be a practical choice for the community modeling, rather than kinetic modeling. However, I think that this does not serve as an right answer to the main point of my comment.

The rationale of FBA is widely known in the research community and it has many known applications. Among possible applications, my point is about the question of how FBA alone can be used to identify competitive or cooperative metabolic interactions and I want to find the solid answer to this point, because this is the rationale of this paper. Specifically, my comment includes the question "In fact, whether the metabolic interactions between two microbial species is competitive or cooperative, it will be dependent on the specific concentrations of metabolites inside the medium, as suggested in many previous FBA studies." The dependency of the nature of metabolic interactions on the metabolite concentration in the media is repeatedly observed in the previous studies of dynamic flux balance analysis (dFBA). If the authors wish to use FBA in their study to claim the identification of competitive or cooperative metabolic interactions, I expect that the authors give the solid justification of how such interactions is identifiable without the consideration of metabolite concentration, while FBA alone does not calculate such concentrations.

2. About my comment about how the authors considered Western diet without the consideration of

the digestion and absorption rate in upper GI tract, the authors answered "Unfortunately it is difficult to find exact uptake rates for each of the modeled metabolites in the small intestine. To circumvent this we depleted all classes of metabolites known to be absorbed in the upper GI tract by tenfold ..."

====> However, although not for all metabolites, there are extensively accumulated knowledge about the digestion/absorption rates of nutrients in the upper GI tract in the available literature, which is not too difficult to find from the Internet. These studies suggest that there are huge variations between the metabolites in their digestion/absorption rates; therefore, I wonder how 10-fold decrease of ALL metabolites in this study can be justifiable as in the authors' claim.

3. My comment includes "it is not yet clear how this specific optimization scheme of Eq. (1) can be rigorously justified. For example, Eq. (1) means the "average" growth rates of microbes. If someone writes ODE of population growth dynamics of the microbial community, it becomes immediately obvious that using of such "average" growth rates as the community-representative quantity is only valid for very short-time periods (shorter than or comparable to doubling times of the fastest-growing microbes), and therefore using of Eq (1) itself sounds rather superficial."

====> The authors answered that fecal data can be viewed as end-point of dynamics and can be viewed as steady-state composition. However, it is still unclear whether this can be an answer to my comment; as I pointed out as above, Eq. (1) can be only used for short-term growth, not for long-term growth dynamics. The authors' answer is red as "opposite" to this point, and I wonder how this can justify the use of Eq. (1).

4. About the use of L2 regularization, the authors wrote some justification in Supplementary Text. However, I want to point out that this argument in the Supplementary Text does not seem to justify their approach. The formulas in Supplementary Text is based on the condition that growth rates are freely adjustable within the conditions imposed in this document. However, please note that the growth rates here are the solutions of FBA models (rather than free variables assumed in standard variational methods) and therefore have already very restricted solution spaces and constraints that are not considered in their argument. Therefore, the conclusions in Supplementary Text seem to involve large mathematical flaws. Note that, resultantly, although the use of such regularization enables the growth of many species as observed in experimental data, only relying on such "empirical" observation cannot constitute mathematical rigor.

5. About validation: It is unclear from their manuscript how they compared the growth rates of individual models to the empirical data provided by Eran Segal's study. The authors claim that they obtained Pearson correlation across and within samples, but specifically, which Pearson correlations do the authors mean? And, which method did they use for P value calculation? Let me assume that the Pearson correlation presented here is the correlation between predicted growth rates of individual species and empirical data. To my knowledge, the Eran Segal's data are about species or strain level. Then, how did the authors reconcile their genus-level result to these data? Also, the low Pearson correlation value (0.2 ~ 0.4) may indicate the correlation made by outlier datapoints, and the authors need to show at least some examples of scatter plots of the data that were used to calculate these Pearson correlations. Also, these low levels of Pearson correlation (0.2~0.4) was achieved by parameter fine-tuning in trade-off and regularization. Can it be really viewed as the validation of the model?

6. About the use of AGORA model concerned by my previous comment, the authors answered that the issue has been resolved in the AGORA authors' reply to the paper that I mentioned. However, even in the AGORA authors' reply, it should be noted that these authors seriously curated their models again in their reply and these new models are available in their website. Therefore, I wonder

the authors of the current manuscript used these new models or erroneous old models. The authors, in their rebuttal letter, claim that "For example, our model results clearly show anaerobic growth of communities across all samples." I really wonder how this crude phenomena can be claimed as a part of a "validation" of their model. The authors also claim that "We do not expect any mechanistic model of a complex microbial community to be completely accurate, and the extent to which the model breaks down or provides non-intuitive results helps us to improve our understanding of the underlying biology." However, I believe that the authors would understand that my comment is not about that the models should be completely accurate; rather, the models should show at least sound mathematical basis and biological accuracy to the level relevant to the study's purpose. For example, are the authors really sure that the old AGORA models are accurate enough to identify cooperative or competitive metabolic interactions as claimed in this study?

7. The English style and figures of the manuscript are still problematic. For example, the Supplementary Text includes at least some typos and weird phrases that made me wonder whether this manuscript was really well proofread by the authors or others.

Reviewer #2 (Comments for the Author):

Diener et al. address many of the concerns of the reviewers including in particular being clearer about the how nutrients available in the gut environment are defined. The authors also now provide some qualitative validation that the growth rates they predict for members of the gut microbial community agree with estimated in vivo replication rates. The modeling method that the authors propose is a useful addition to methods that are used to analyze complex microbial communities based on limited data (only abundance and rough nutritional information). Much further experimental validation would be needed to figure out if this particular modeling method can provide reliable quantitative predictions of nutrient exchanges in complex microbial communities.

Major comments:

1) As I stated in my review of the first version of the manuscript, I would have liked to see a validation of the proposed computational method with a smaller well-characterized community (like some of the industrial communities that have been studied previously). The paper is ok without this, but the method should be considered to be only partially quantitative due to lack of validation against real quantitative data in a more controlled setting. The authors should be explicit about this being a qualitative model of microbial communities and being limited by the availability of more quantitative data than species abundance distributions.

2) The authors mention that their constraints are the same that are used in the SteadyCom method from the Maranas group. However, they do not specify how MiCom is different from SteadyCom explicitly. The major difference is that SteadyCom (correctly) requires that all members of the community grow at the same growth rate over relevantly long time periods (i.e. that no single organism takes over). In MiCom the growth rates can vary, but L2 regularization is used to make sure that non-sparse solutions where all observed species can grow are favored. I would like to see a clear discussion of MiCom differences with SteadyCom, and ideally also a comparison of nutrient exchange or species knockout predictions between MiCom and SteadyCom.

Minor comments:

1) L84: I think you mean open source here

2) L215: How sensitive are model predictions to the exact factor used here (e.g. 10 vs 20)?

3) L240: This paragraph alternates between specifying the tradeoff parameter in %'s and in fractions. Please make this consistent.

4) Figure 4B: I'm not clear of what the purpose of this figure is.

Response to reviewers (second revision)

All mentioned line numbers refer to the **marked-up manuscript.**

Reviewer #1 (Comments for the Author)

In this revision, the authors have addressed several issues raised by the reviewers' comments and I appreciate their efforts. However, I think that the revision is far from satisfactory and it detours the critical points in my comments. Also, the validation of the model is not yet clear, and the mathematical ground of their method, for example the regularization part in Supplementary Text, seems to involve mathematical flaws.

1. My previous comments include "however, I wonder how such static scheme actually predicts the nature of metabolic interactions between microbes (which is major claim of this paper). In fact, whether the metabolic interactions between two microbial species is competitive or cooperative, it will be dependent on the specific concentrations of metabolites inside the medium, as suggested in many previous FBA studies. With such a static scheme in this paper, it is not mathematically feasible to calculate such concentration of metabolites, and therefore it is hard to accept the main claim of this paper on the nature of metabolic interactions between microbial species."

==> In the authors' rebuttal letter, I do not think that the authors gave a satisfactory answer to this major comment. To my best interpretation, the authors gave only the justification of why FBA can be a practical choice for the community modeling, rather than kinetic modeling. However, I think that this does not serve as an right answer to the main point of my comment. The rationale of FBA is widely known in the research community and it has many known applications. Among possible applications, my point is about the question of how FBA alone can be used to identify competitive or cooperative metabolic interactions and I want to find the solid answer to this point, because this is the rationale of this paper. Specifically, my comment includes the question "In fact, whether the metabolic interactions between two microbial species is competitive or cooperative, it will be dependent on the specific concentrations of metabolites inside the medium, as suggested in many previous FBA studies." The dependency of the nature of metabolic interactions on the metabolite concentration in the media is repeatedly observed in the previous studies of dynamic flux balance analysis (dFBA). If the authors wish to use FBA in their study to claim the identification of competitive or cooperative metabolic interactions, I expect that the authors give the solid justification of how such interactions is identifiable without the consideration of metabolite concentration, while FBA alone does not calculate such concentrations.

The feasibility of FBA to identify cooperative or competitive behavior has been established. For instance, see work by Zomorodi and Maranas (<https://doi.org/10.1371/journal.pcbi.1002363>), Mendes-Soares et. al.

(<https://bmcbioinformatics.biomedcentral.com/articles/10.1186/s12859-016-1230-3>), and this is reviewed in <https://doi.org/10.6084/m9.figshare.c.4695341>. It is true that the absence or presence of a metabolite in the external medium can alter cooperation or competition between taxa, however this would require uptake or production of this metabolite by competing or cooperating taxa, and thus does depend on fluxes rather than just concentrations. FBA assumes steady-state conditions for metabolite concentration and fluxes. FBA provides steady state metabolite fluxes, which in our view are the consequence of metabolite-mediated interactions between taxa.

2. About my comment about how the authors considered Western diet without the consideration of the digestion and absorption rate in upper GI tract, the authors answered "Unfortunately it is difficult to find exact uptake rates for each of the modeled metabolites in the small intestine. To circumvent this we depleted all classes of metabolites known to be absorbed in the upper GI tract by tenfold ..."

====> However, although not for all metabolites, there are extensively accumulated knowledge about the digestion/absorption rates of nutrients in the upper GI tract in the available literature, which is not too difficult to find from the Internet. These studies suggest that there are huge variations between the metabolites in their digestion/absorption rates; therefore, I wonder how 10-fold decrease of ALL metabolites in this study can be justifiable as in the authors' claim.

We agree with the reviewer that there is some existing knowledge on which general classes of metabolites are absorbed in the small intestine. This knowledge was used to decrease the maximum import fluxes *for only those compounds that are known to be absorbed in the small intestine* (<https://dx.doi.org/10.1016%2Fj.bpg.2016.02.007>). The statement that we performed a "10-fold decrease of ALL metabolites in this study" is not accurate. Our manuscript states that "To account for uptake in the small intestine, we reduced all import fluxes for metabolites commonly absorbed in the small intestine by a factor of 10." This seemed to be the best we could do with the knowledge that we currently have. The list of these metabolites that were depleted from the model is available in the associated Github repository at https://github.com/micom-dev/paper/blob/master/data/western_diet.csv. However, if the reviewer is aware of any publication that reports a comprehensive list of empirically derived import fluxes for every metabolite within the human gut (i.e. *in vivo*) we would appreciate it if they could provide us with the reference.

3. My comment includes "it is not yet clear how this specific optimization scheme of Eq. (1) can be rigorously justified. For example, Eq. (1) means the "average" growth rates of microbes. If someone writes ODE of population growth dynamics of the microbial community, it becomes immediately obvious that using of such "average" growth rates as the community-representative quantity is only valid for very short-time periods (shorter than or comparable to doubling times of the fastest-growing microbes), and therefore using of Eq (1) itself sounds rather superficial."

====> The authors answered that fecal data can be viewed as end-point of dynamics and can be viewed as steady-state composition. However, it is still unclear whether this can be an answer to my comment; as I pointed out as above, Eq. (1) can be only used for short-term growth, not for long-term growth dynamics. The authors' answer is red as "opposite" to this point, and I wonder how this can justify the use of Eq. (1).

Equation 1 simply states the overall amount of biomass produced in the system and is valid as long as the abundances and growth rates for individual taxa are approximately constant (i.e. steady state assumption). We clarify that Equation 1 does not represent an "average" -- it represents an effective biomass production of the microbial community. As the equation indicates, the effective biomass intuitively depends on abundances and metabolic capacities in the microbial community at steady-state. There is no 'long-term' or 'short-term' under this assumption -- a steady state implies constant, chemostat-like conditions. We provided references in the previous revision indicating that this assumption seems reasonable in the gut and also stated this assumption explicitly in the manuscript (lines 109-115).

4. About the use of L2 regularization, the authors wrote some justification in Supplementary Text. However, I want to point out that this argument in the Supplementary Text does not seem to justify their approach. The formulas in Supplementary Text is based on the condition that growth rates are freely adjustable within the conditions imposed in this document. However, please note that the growth rates here are the solutions of FBA models (rather than free variables assumed in standard variational methods) and therefore have already very restricted solution spaces and constraints that are not considered in their argument. Therefore, the conclusions in Supplementary Text seem to involve large mathematical flaws. Note that, resultantly, although the use of such regularization enables the growth of many species as observed in experimental data, only relying on such "empirical" observation cannot constitute mathematical rigor.

We apologize for the confusion. We have now expanded the section stating our assumptions in the derivation (see Supplemental Text S1). We agree that the presented solution to Equation 2 depends on the imposed feasibility constraints, which are not included in that approximation. In light of this, we performed an additional analysis in the revised manuscript that specifically investigates how well the derived expression captures the numerical solution, which does consider all constraints (lines 271-281). As we have shown, the derived expression coincides with the full solution (*considering all imposed constraints and feasibility*) with an average R^2 of 0.94 and usually describes the full solution very well as shown in Figure 3A.

5. About validation: It is unclear from their manuscript how they compared the growth rates of individual models to the empirical data provided by Eran Segal's study. The authors claim that they obtained Pearson correlation across and within samples, but specifically, which Pearson correlations do the authors mean? And, which method did they use for P value calculation? Let me assume that the Pearson correlation presented here is the correlation between predicted growth rates of individual species and empirical data. To my knowledge, the Eran Segal's data

are about species or strain level. Then, how did the authors reconcile their genus-level result to these data? Also, the low Pearson correlation value (0.2 ~ 0.4) may indicate the correlation made by outlier datapoints, and the authors need to show at least some examples of scatter plots of the data that were used to calculate these Pearson correlations. Also, these low levels of Pearson correlation (0.2~0.4) was achieved by parameter fine-tuning in trade-off and regularization. Can it be really viewed as the validation of the model?

We apologize for the lack of clarity here and have now included an additional section on how those correlations were obtained in the Methods section (lines 519-524). Replication rates were indeed obtained at the strain level and summarized to the mean within the respective taxonomic level used in the MICOM community models (genus or species). This is now stated in lines 228-230. We now also include examples of correlations between the replication rates and predicted growth rates in Fig. S1. Additionally, running the analysis with Spearman rank correlations (which should be robust against outliers) gave similar results to Pearson correlation (see lines 246-250, Spearman $r = 0.28$ and 0.27 within and across samples, respectively).

6. About the use of AGORA model concerned by my previous comment, the authors answered that the issue has been resolved in the AGORA authors' reply to the paper that I mentioned. However, even in the AGORA authors' reply, it should be noted that these authors seriously curated their models again in their reply and these new models are available in their website. Therefore, I wonder the authors of the current manuscript used these new models or erroneous old models. The authors, in their rebuttal letter, claim that "For example, our model results clearly show anaerobic growth of communities across all samples." I really wonder how this crude phenomena can be claimed as a part of a "validation" of their model. The authors also claim that "We do not expect any mechanistic model of a complex microbial community to be completely accurate, and the extent to which the model breaks down or provides non-intuitive results helps us to improve our understanding of the underlying biology." However, I believe that the authors would understand that my comment is not about that the models should be completely accurate; rather, the models should show at least sound mathematical basis and biological accuracy to the level relevant to the study's purpose. For example, are the authors really sure that the old AGORA models are accurate enough to identify cooperative or competitive metabolic interactions as claimed in this study?

We apologize for not clearly stating the version of AGORA reconstructions that was used. The initial submission used version 1.01 of AGORA, which included the fixes mentioned by the reviewer. The current manuscript is based on version 1.03, which is now clearly stated in the text at lines 196, 515 and 664. Based on prior work with these models we feel that they are sufficiently accurate for the purposes of the work presented here.

7. The English style and figures of the manuscript are still problematic. For example, the Supplementary Text includes at least some typos and weird phrases that made me wonder whether this manuscript was really well proofread by the authors or others.

We have made a concerted effort to polish our prose throughout the manuscript, and in particular in the supplemental material and figure captions.

Reviewer #2 (Comments for the Author):

Diener et al. address many of the concerns of the reviewers including in particular being clearer about the how nutrients available in the gut environment are defined. The authors also now provide some qualitative validation that the growth rates they predict for members of the gut microbial community agree with estimated *in vivo* replication rates. The modeling method that the authors propose is a useful addition to methods that are used to analyze complex microbial communities based on limited data (only abundance and rough nutritional information). Much further experimental validation would be needed to figure out if this particular modeling method can provide reliable quantitative predictions of nutrient exchanges in complex microbial communities.

We thank the reviewer for their positive evaluation of the manuscript.

Major comments:

1) As I stated in my review of the first version of the manuscript, I would have liked to see a validation of the proposed computational method with a smaller well-characterized community (like some of the industrial communities that have been studied previously). The paper is ok without this, but the method should be considered to be only partially quantitative due to lack of validation against real quantitative data in a more controlled setting. The authors should be explicit about this being a qualitative model of microbial communities and being limited by the availability of more quantitative data than species abundance distributions.

We agree with the reviewer in that fully quantitative community models would require additional validation with fluxomics data. Unfortunately only a few studies report data with the resolution that is required to compare it to MICOM predictions. For the few studies we identified that may be applicable, the required data was not included in the manuscript and authors did not reply to inquiries. We hope to generate validation data sets in the future. In the meantime, we have added phrasing that clearly states that the metabolic interactions inferred by MICOM should be seen as qualitative and will require further validation work (lines 33, 453-454). In terms of growth rate predictions we do feel that MICOM is roughly quantitative. The significant associations between MICOM growth rate predictions and *in vivo* replication rates is quantitatively demonstrated (i.e. Spearman and Pearson correlations).

2) The authors mention that their constraints are the same that are used in the SteadyCom method from the Maranas group. However, they do not specify how MiCom is different from SteadyCom explicitly. The major difference is that SteadyCom (correctly) requires that all members of the community grow at the same growth rate over relevantly long time periods (i.e.

that no single organism takes over). In MiCom the growth rates can vary, but L2 regularization is used to make sure that non-sparse solutions where all observed species can grow are favored. I would like to see a clear discussion of MiCom differences with SteadyCom, and ideally also a comparison of nutrient exchange or species knockout predictions between MiCom and SteadyCom.

We have now included an additional section comparing SteadyCom and MICOM (lines 559-565). Even though both methods use a similar set of constraints they differ greatly in their goals and assumptions. SteadyCom is used to predict abundances from a list of candidate taxa, assuming the same dilution and growth rates for each taxon. MICOM is used to predict growth rates and exchange fluxes from a list of taxon abundances and does not assume the same dilution and growth rate for each taxon. Even though SteadyCom's assumption of the same growth rate for each taxon does simplify calculations, we do not feel that it is backed by empirical evidence. For example, growth rates appear to vary greatly between taxa in the human gut, as was first shown by Korem et al. (<https://doi.org/10.1126/science.aac4812>) and has been confirmed by a handful of other studies.

Minor comments:

- 1) L84: I think you mean open source here
- 2) L215: How sensitive are model predictions to the exact factor used here (e.g. 10 vs 20)?
- 3) L240: This paragraph alternates between specifying the tradeoff parameter in %'s and in fractions. Please make this consistent.
- 4) Figure 4B: I'm not clear of what the purpose of this figure is.

- 1) This has been replaced with "open-source".
- 2) We have included an analysis of varying that parameter in Figure S4 which is also referenced in the main text at lines 334-336.
- 3) Tradeoff is now consistently specified as fractions.
- 4) An additional explanation has now been added to the figure legend (lines 1110-1111).

December 19, 2019

Dr. Osbaldo Resendis-Antonio
Coordinacion de la Investigacion Cientifica - RAI UNAM
Mexico City
Mexico

Re: mSystems00606-19R1 (MICOM: metagenome-scale modeling to infer metabolic interactions in the gut microbiota)

Dear Dr. Osbaldo Resendis-Antonio:

Your manuscript has been accepted, and I am forwarding it to the ASM Journals Department for publication. For your reference, ASM Journals' address is given below. Before it can be scheduled for publication, your manuscript will be checked by the mSystems production editor, Ellie Ghatineh, to make sure that all elements meet the technical requirements for publication. She will contact you if anything needs to be revised before copyediting and production can begin. Otherwise, you will be notified when your proofs are ready to be viewed.

Sincerely,

Nicholas Chia
Editor, mSystems

Journals Department
Supplemental Figure S5: Accept
Supplemental Figure S3: Accept
Supplemental Figure S2: Accept
Supplementary Text S1: Accept
Supplemental Figure S4: Accept
Supplemental Figure S1: Accept